# Prognostic Impact of CD38- and IgκC-Positive Tumor-Infiltrating Plasma Cells in Triple-Negative Breast Cancer

**DOI:** 10.3390/ijms242015219

**Published:** 2023-10-16

**Authors:** Anne-Sophie Heimes, Natali Riedel, Katrin Almstedt, Slavomir Krajnak, Roxana Schwab, Kathrin Stewen, Antje Lebrecht, Marco Johannes Battista, Walburgis Brenner, Annette Hasenburg, Marcus Schmidt

**Affiliations:** Department of Obstetrics and Gynecology, University Medical Center, Johannes Gutenberg University Mainz, 55131 Mainz, Germany; natalie.riedel@freenet.de (N.R.); katrin.almstedt@unimedizin-mainz.de (K.A.); slavomir.krajnak@unimedizin-mainz.de (S.K.); roxana.schwab@unimedizin-mainz.de (R.S.); kathrin.stewen@unimedizin-mainz.de (K.S.); antje.lebrecht@unimedizin-mainz.de (A.L.); marco.battista@unimedizin-mainz.de (M.J.B.); brenner@uni-mainz.de (W.B.); annette.hasenburg@unimedizin-mainz.de (A.H.); marcus.schmidt@unimedizin-mainz.de (M.S.)

**Keywords:** triple-negative breast cancer (TNBC), tumor-infiltrating plasma cells, CD38, IgκC

## Abstract

Due to a higher mutational load, triple-negative breast cancer (TNBC) is characterized by a higher immunogenicity compared to other subtypes. In this context, we analyzed the prognostic significance of tumor-infiltrating plasma cells in a cohort of 107 triple-negative breast cancer patients. Tumor-infiltrating plasma cells were analyzed via immunohistochemistry using the plasma cell markers CD38 and IgκC. The prognostic impact of the CD38 and IgκC expression was evaluated using the Kaplan–Meier plots and Cox regression analyses. A Spearman-Rho correlation coefficient was used to evaluate a possible association between plasma cell infiltration and the *BRCA* mutation status. The study cohort consisted of 107 patients with early-stage TNBC, who were treated between 2009 and 2016 at the Department of Gynecology and Obstetrics, University Medical Center Mainz, Germany. The median follow-up was five years. The Kaplan–Meier survival analysis showed that higher tumor infiltration with CD38-positive plasma cells was associated with significantly longer metastasis-free survival (MFS) (*p* = 0.039 Log Rank). In the multivariate Cox regression analysis for metastasis-free survival, in which additional clinicopathological factors (age, tumor size, nodal status, and grading) were considered, CD38 was identified as an independent prognostic factor within the analyzed cohort (HR 0.438, 95% CI 0.195–0.983; *p* = 0.045). In addition to the CD38 expression, the nodal status was also identified as an independent prognostic factor in multivariate Cox regression. Regarding the IgκC expression, a higher IgκC expression was shown to be associated with a better outcome, although this effect was not statistically significant. Furthermore, we were able to show a significant correlation between plasma cell infiltration and the *BRCA* mutation status. A favorable prognostic significance of tumor-infiltrating plasma cells could be demonstrated in triple-negative breast cancer immunohistochemically analyzed for the CD38 and IgκC expression. CD38 was identified as an independent prognostic factor via multivariate Cox regression.

## 1. Introduction

Triple-negative breast cancer, which is defined by a lack of expression of the estrogen, progesterone, and human epidermal growth factor receptor 2 (HER2) receptors, represents approximately 10–15% of all breast cancers and is characterized by more aggressive tumor growth, poorer differentiation, and a higher proliferation index (Ki-67), as well as the associated unfavorable prognosis [1]. The immune system plays an increasingly important role in the development of targeted therapeutic strategies in breast cancer. Using mRNA sequencing data from The Cancer Genome Atlas (TCGA), it has been shown that a high expression of T-cell and B-cell signatures has a favorable effect on the overall survival in many tumor entities, including breast cancer [2]. Due to a higher mutational load, and consequently, a higher number of immunogenic tumor-associated antigens, triple-negative breast cancer has a stronger immunogenic potential compared to the other subtypes [3]. Tumor-associated antigens (TAA) can be recognized by the immune system and elicit an immune response that, in the best case, leads to tumor suppression [4]. In this context, tumor-infiltrating lymphocytes (TILs) play an important role: numerous studies have already shown that TILs have a strong prognostic and predictive influence in triple-negative breast cancer [5,6,7,8,9,10,11], with higher numbers of tumor-infiltrating lymphocytes being associated with a higher rate of pathologic complete remission (pCR) and improved survival. In a recent retrospective analysis by de Jong et al. in which a total of 441 younger patients with early node-negative TNBC who did not receive (neo)adjuvant systemic therapy were included, the presence of stromal TILs (sTILs) was identified as an independent prognostic parameter; an increased number of stromal TILs was associated with longer MFS and OS [12]. Based on the excellent prognosis of those younger patients (age at diagnosis <40) with early node-negative TNBC, small tumor size, and a high level of sTILs, the authors postulated that de-escalating strategies should be evaluated in prospective clinical trials, demonstrating a possible clinical impact of sTILs in early triple-negative breast cancer. 

Stanton and colleagues showed in a systematic review that the extent of tumor-infiltrating lymphocytes varies both within and between subtypes, with triple-negative breast cancer often having high levels of TILs [13]. TILs are composed of various immune cells (T-lymphocytes, B-lymphocytes, and macrophages) that interact with cytokines, immunoglobulins, interferons, and other immune-associated mediators to form the so-called tumor microenvironment. Following the presentation of antigens by antigen-presenting cells (APCs), various immune responses can be elicited. Depending on the predominant immune cell type, either a tumor-inhibiting effect (acute inflammation by CD8 lymphocytes (Th1) or M1 macrophages) or a Th2-driven tumor-promoting effect (chronic inflammation by M2 macrophages, regulatory T cells, or immune checkpoints such as programmed cell death protein 1 (PD-1) or its ligand (PD-L1) can result. The complex interaction between the immune system and the tumor is illustrated via the so-called “Cancer Immunity Cycle” [4]. Most studies analyzing the prognostic and/or predictive role of TILs in breast cancer have focused on the cellular immune system, particularly cytotoxic T-cells. For example, several studies have demonstrated that increased numbers of tumor-infiltrating lymphocytes or T-cell transcripts are associated with improved prognosis in rapidly proliferating breast cancers, such as triple-negative breast cancer [14,15,16]. Recently, immune checkpoint inhibitors in combination with chemotherapy have been successfully used in the treatment of early and advanced triple-negative breast cancer, which release the “brakes” on the immune system and enhance naturally occurring immune responses against the tumor [17,18,19]. In contrast, the role of B lymphocytes, especially tumor-infiltrating plasma cells, is controversial [20]. Using the gene expression analysis, we and others have shown that the expression of B-cell-associated genes (collectively referred to as B-cell metagene) is associated with a favorable prognosis in patients with early node-negative breast cancer [21,22,23]. Further studies have confirmed the prognostic significance of IgκC as a representative marker of the B-cell metagene at both the mRNA and protein levels [24,25]. Furthermore, IgκC has been identified as a predictive marker for response to anthracycline-based chemotherapy [24]. Using fluorescence microscopy, IgκC was detected in IRF4-positive tumor-infiltrating plasma cells [24]. The primary objective of the current study was to evaluate the prognostic impact of tumor-infiltrating plasma cells in a cohort of 107 patients with early triple-negative breast cancer. The expression of the plasma cell markers CD38 and IgκC, was determined via immunohistochemistry. Additionally, we analyzed whether there was a correlation between plasma cell infiltration (the expression of CD38 and IgκC) and the histological degree of differentiation and the proliferation Index Ki67, respectively. Other secondary objectives were to determine whether the expression levels of CD38 and IgκC correlated with each other and to evaluate the extent to which there was a correlation between the *BRCA* mutation (at the germline level) and the level of plasma cell infiltration.

## 2. Results

To evaluate the impact of plasma cell infiltration in the below-mentioned cohort of triple-negative breast cancer patients, the plasma cell markers CD38 and IgκC were analyzed via immunohistochemistry using a semi-quantitative scoring method as previously described in [25,26]. To dichotomize the patients, tumor slides with IgκC/CD38 score 0 and 1+ were considered as low plasma cell infiltration, whereas cases with 2+ and 3+ were high plasma cell infiltration, respectively.

The immunohistochemical examination of tumor slides showed 55.1% of tumor slides as weakly positive or negative for the IgκC expression (Score 0, 1+), with 44.9% being strongly IgκC positive (Score 2+, 3+). Concerning the CD38 expression, 56.1% of the tumor slides were weakly positive or negative, whereas 43.9% showed a strong CD38 expression (Score 2+, 3+).

### 2.1. Prognostic Impact of IgκC Expression

The Kaplan–Meier analysis demonstrated that a higher IgκC expression had a weak trend to be associated with a better outcome (longer MFS), although this effect did not reach the statistical significance level (*p* = 0.35 Log Rank) (Figure 1).

The univariate and multivariate Cox Regression analysis, adjusted for the clinical–pathological parameters such as age, tumor size, lymph node status, and histological grade of differentiation, failed to show a statistically significant impact of the IgκC expression on metastasis-free survival in the observed cohort (Table 1 and Table 2). However, the lymph node status was shown to be a significant prognostic factor in both the univariate and multivariate Cox regression analysis.

### 2.2. Prognostic Impact of CD38 Expression

Using the Kaplan–Meier survival analysis, it was demonstrated that a higher number of CD38-positive tumor-infiltrating plasma cells was associated with a significantly longer metastasis-free survival within the observed collective (*p* = 0.039 Log Rank) (Figure 2). In the multivariate Cox regression analysis adjusted for other clinicopathological factors (age, tumor size, nodal status, and grading), CD38 was identified as an independent prognostic factor within the analyzed collective (HR 0.438, 95% CI 0.195–0.983; *p* = 0.045). In addition to the CD38 expression, the nodal status was also shown to be an independent prognostic factor in multivariate Cox regression (Table 3). 

### 2.3. Validation of the Prognostic Impact of IgκC and CD38 Expression Using Gene Expression Data from an Independent, Publicly Available TNBC Cohort

To validate our immunohistochemical results on the prognostic impact of the plasma cell markers IgκC and CD38 in a larger, independent cohort of TNBC, we used the publicly available gene expression data of IgκC and CD38 with their associated tumor characteristics and follow-up data [27] in a cohort of 424 TNBC samples. Using two different probe sets for the IgκC mRNA expression, the Kaplan–Meier plots show that a higher IgκC mRNA expression was associated with a significantly longer MFS (probe set 214669_x_at: *p* = 0.049 Log Rank; probe set 216576_x_at: *p* = 0.018 Log Rank, Figure 3). These validation results on the mRNA gene expression data of IgκC are basically in line with our immunohistochemical results showing a trend that a higher IgκC expression at the protein level correlates with a longer MFS. With regard to CD38, the mRNA expression data do not show a significant effect on MFS (Figure 4).

### 2.4. Correlation between the Strength of Plasma Cell Infiltration (CD38 and IgκC Expression) and the Tumor Grade as Well as the Proliferation Index Ki67

In addition, we investigated whether there was a correlation between the IgκC and CD38 expression on the one hand and the histological degree of differentiation and the proliferation index Ki67 on the other hand. Using ordinal regression, a significant correlation between the CD38 expression and the proliferation index Ki67 was found: a higher CD38 expression was associated with a higher proliferation index (*p* = 0.024).

In contrast, there was no significant correlation between IgκC and the proliferation index Ki67 or the histological grade of differentiation, nor between the CD38 expression and the histological grade of differentiation. 

### 2.5. Correlation of the Level of Tumor-Infiltrating Plasma Cells and BRCA Mutation Status

Using the Spearman-Rho correlation, we could show a strong, statistically significant correlation between the level of the CD38 expression and the level of the IgκC expression (r = 0.588, *p* < 0.001). In a subsequent step, we were able to demonstrate a significant correlation between the presence of a BRCA mutation (at the germline level) and the extent of plasma cell infiltration (determined as the mean of the IgκC and CD38 expression) using the Spearman-Rho correlation coefficient: the presence of a BRCA mutation was associated with a stronger plasma cell infiltration (r = 0.211; *p* = 0.029) (Figure 5).

## 3. Discussion

The present retrospective study demonstrated a favorable prognostic impact of tumor-infiltrating plasma cells within the described collective of 107 triple-negative breast cancer patients, which were analyzed immunohistochemically with regard to the expression of the plasma cell markers CD38 and IgκC. Furthermore, CD38 was identified as an independent prognostic factor via multivariate Cox regression. Additionally, we could show a significant correlation between the CD38 expression and the proliferation index Ki67: a higher number of CD38-positive, tumor-infiltrating plasma cells was associated with a higher percentage of Ki67-positive tumor cells. 

These results confirmed the prognostic significance of IgκC as a representative marker of the B-cell metagene and underlined that the humoral immune response plays an important role in antitumor immunity. These findings are in line with previous studies of our group and others. Refs. [21,22,23,24,25,28]: using the gene expression analysis, IgκC has already been shown to be a prognostic marker in a large cohort consisting of 965 node-negative breast cancer patients who did not receive any systemic therapy in the adjuvant setting [24], where a higher IgκC mRNA expression was associated with a better prognosis [24]. This effect was particularly pronounced in patients with estrogen receptor (ER)-negative, highly proliferating breast cancer. Additionally, IgκC was identified as a predictive marker for the response to neoadjuvant anthracycline-based chemotherapy in a cohort of 845 breast cancer patients, where the higher IgκC expression was correlated with a higher pCR rate. Further immunohistochemical studies confirmed that IgκC is mainly produced by activated tumor-infiltrating plasma cells. In a retrospective immunohistochemical study of Chen et al., the prognostic impact of the IgκC expression was evaluated in a cohort of 335 node-negative breast cancer patients who did not receive any adjuvant therapy after operation and irradiation [25]. Of these, 160 patients had infiltration with IgκC-positive, tumor-infiltrating plasma cells, which showed a statistically significant longer survival time. In estrogen receptor-negative tumors and in luminal B breast carcinomas, the significance was particularly emphasized. In the present work, adjuvant chemotherapy could be a possible bias for the evaluation of the prognosis. In contrast, regarding the above-mentioned study from Schmidt et al. [24] and the immunohistochemical study of Chen et al. [25], only patients who did not receive any adjuvant systemic treatment entered their studies, allowing for the validation of IgκC as a possible prognostic factor. Furthermore, the above-cited studies included all breast cancer subtypes. In contrast, the present study focused on triple-negative breast cancer samples. In our retrospective immunohistochemical study, there was a trend for longer metastasis-free survival in tumors with a higher IgκC expression. In contrast to the above-mentioned studies, statistical significance was not reached. This could be due to the small number of cases or the restriction to triple-negative breast cancer or the chemotherapy administered. Contrasting these findings, another retrospective study analyzed the prognostic significance of IgκC, whose expression was evaluated at the protein level using immunohistochemistry [26]. Thereby, a significant prognostic influence of the IgκC expression could be confirmed even in patients treated with adjuvant chemotherapy. A higher IgκC expression was associated with longer metastasis-free survival. These retrospective results regarding the prognostic significance of the IgκC expression in adjuvant chemotherapy-treated patients with early breast cancer were confirmed using the retrospective–prospective data. The importance of the IgκC gene expression as a positive prognostic marker was demonstrated within a large cohort of patients with early breast cancer treated in the prospective randomized FinHer trial [29]. Thus, it was shown that a higher IgκC expression was associated with longer metastasis-free survival, particularly in patients with triple-negative breast cancer. 

These results are basically in line with the findings of our present retrospective immunohistochemical study which included 107 patients with early triple-negative breast cancer, from whom 79 received an adjuvant chemotherapy, showing a trend for a higher IgκC expression being associated with a better clinical outcome. These immunohistochemical results are supported by the validation of the IgκC expression using publicly available mRNA data from a larger, independent TNBC cohort: here, two probe sets were able to demonstrate that a higher IgκC expression was associated with significantly longer MFS (Figure 3). 

Furthermore, in the present study, we demonstrated a prognostic effect of CD38-positive, tumor-infiltrating plasma cells, where a higher expression of CD38 was associated with a longer MFS. Additionally, CD38 could be identified as a positive independent prognostic marker (using the multivariate Cox Regression analyses). These results could be confirmed for relapse-free survival (RFS) and disease-free survival (DFS) by other studies, i.e., the plasma cell marker CD38, among other markers of tumor-infiltrating lymphocytes, was used in a Japanese retrospective study including 114 patients with triple-negative breast cancer. The authors postulated a significant prognostically favorable impact of CD38-positive, tumor-infiltrating plasma cells for relapse-free survival [28]. These findings are in line with the results of a retrospective study published in 2018 by Yeong et al. [30]. Using immunohistochemistry, the authors evaluated the prognostic impact of plasma cell-associated markers such as CD38 in a cohort of 269 TNBC samples. They revealed a statistically significant longer disease-free survival (DFS) in patients with a higher density of CD38-positive plasma cells (*p* = 0.004). Furthermore, they could show a significant correlation between the amount of CD38-positive plasma cells and the expression of B-cell-associated genes such as IgκC, IGHM (Immunoglobulin Heavy Constant Mu), and IGHG (Immunoglobin Heavy Constant Gamma) [30]. 

Antitumor immunity represents an important mechanism in tumor biology in general, not only in breast cancer but also in other tumor entities. Schmidt et al. were able to show that the humoral immune response has a favorable prognostic effect in non-small cell lung cancer (NSCLC) [24]. In a cohort of 196 NSCLC patients, the gene expression analyses revealed that the upregulation of the B-cell metagene and a higher IgκC expression were associated with longer survival. Within the subgroup analysis, this effect was significant only in adenocarcinomas, but not in squamous cell carcinomas. In addition, a retrospective Japanese study could show that the presence of TILs also has a favorable prognostic effect in endometrial carcinoma [31]. Based on the immunohistochemical analyses, it could be demonstrated that the B-cell infiltration (measured via the immunohistochemical expression of CD20 and CD38) correlated positively with the number of TILs and was associated with a better outcome.

The key secondary objective of our study was to examine whether a BRCA mutation was associated with a higher rate of tumor-infiltrating plasma cells. We showed a positive correlation of tumor-infiltrating plasma cells and mutations in BRCA. This finding may be explained via genetic instability and the resulting higher number of immunogenic antigens in the tumor, especially in the presence of a BRCA mutation [32]. The use of PARPi is known to increase the genomic instability already present in the tumor, so treatment with PARPi (poly ADP-ribose polymerase inhibitor) may increase the sensitivity to immunotherapy in BRCAm tumors. In a retrospective study, Grandal et al. showed that in patients with luminal breast cancer, the BRCA mutation status was associated with higher pCR rates and higher numbers of TILs after neoadjuvant chemotherapy [33], but they did not find a correlation between immune infiltration at the baseline and the BRCA mutation status. Another study by Solinas et al. demonstrated that there was a significantly higher rate of TIL-positive tumors in the BRCA-mutated (BRCAm) TNBC group compared to the BRCA-wildtype (wt) group (*p* = 0.037) [34]. However, the study cited above did not find a significant difference in the composition of TILs in the BRCAm TNBC group compared to the BRCAwt group. A further study by Telli et al. [35] evaluated the correlation between TILs and the BRCA mutation status in a cohort of 161 TNBC patients pooled from five phase II clinical trials of platinum-based neoadjuvant therapy. They did not find a significant association between TILs and the BRCA mutation status in the observed cohort of TNBC patients. Using TCGA data, Kraya et al. were able to show that BRCA-mutated breast cancer tumors dispose of a higher immunogenicity compared with those without mutation. In contrast, they found a negative correlation between HRD (Homologous recombination deficiency) scores and the gene expression-based immune markers [36]. The above-mentioned studies show that, in conclusion, the association between immune infiltration and BRCA mutation in triple-negative breast cancer is not yet clearly established and requires further studies with larger sample sizes.

The strength of our retrospective study is that we showed a positive prognostic impact of CD38- and IgκC-positive tumor-infiltrating plasma cells in TNBC patients. The differences in prognostic significance between the protein and the gene expression could be explained via post-translational changes and the different sample sizes analyzed for immunohistochemistry and the gene expression, respectively. Furthermore, a correlation between tumor-infiltrating plasma cells and BRCA mutations could be demonstrated. Potential weaknesses of the present study include the small sample size of 107 tumors, differences in systemic treatment (chemotherapy vs. no systemic therapy), differences in the analysis methods and sample size between the finding and validation cohorts (immunohistochemistry vs. gene expression), and the retrospective study design. 

A further potential weakness of the retrospective study presented here is that the prognostic significance of stromal TILs was not analyzed, assuming the fact that the significance of stromal TILs in early triple-negative breast cancer is sufficiently well known [5,6,7,8,9,10,11,12]. However, it is not possible to conclude from this study whether the amount of plasma cell infiltration provides additional prognostic information compared to stromal TILs.

## 4. Material and Methods

### 4.1. Study Patients

The patient collective consisted of 107 patients who had been treated for triple-negative breast cancer between 2009 and 2016 at University Hospital Mainz and received breast-conserving surgery or mastectomy. Following surgical treatment, 79 patients received chemotherapy and 28 of the 107 patients were without chemotherapy, although there is a general recommendation for (neo)adjuvant chemotherapy in triple-negative tumors upon a tumor size pT1b and larger. According to the results of the follow up questionnaire and our database, the most common reasons for this were general refusal or the discontinuation of chemotherapy in the case of intolerance due to side effects. At the time of the diagnosis, the median age was 55 years. The proportion of BRCA mutation-associated triple-negative breast cancers was 15.9%, including 14.02% with BRCA-1 mutation and 1.88% with BRCA-2 mutation. The median follow up was five years. 

At the time of the last follow-up, 88.8% of the patients were without metastases, whereas 11.2% developed distant metastases. The recurrence of their diseases was reported by 3.7% of the patients; thus, 96.3% were without recurrence.

To perform the immunohistochemistry of CD 38 and IgκC, tissue samples were available from the whole study cohort of 107 patients. Tumor slides were provided by the tissue bank of the University Medical Center Mainz in accordance with the regulations of the tissue biobank and the approval of the ethics committee of University Medical Center Mainz.

Patients’ characteristics are given in Table 4.

### 4.2. Immunostaining

For immunostaining, formalin-fixed and paraffin-embedded tumor slides were stained with commercially available monoclonal antibodies binding IgκC (Clone KP-53; Santa Cruz Biotechnology Company, Santa Cruz, CA, USA) and CD38 (Sigma-Aldrich Chemie GmbH, Schnelldorf, Germany), according to the standard procedures as previously described [26]. All slides were analyzed using a Leica light microscope (Leica Microsystem Vertrieb Company, Wetzlar, Germany) by three of the authors (N.R., A.-S.H., and M.S.) trained in histological and immunohistochemical diagnostics, unaware of the clinical outcome.

### 4.3. Evaluation of Immunostaining

For the immunohistochemical evaluation of IgκC- and/or CD38-positive tumor-infiltrating plasma cells, a semiquantitative scoring method was used as previously described [25]: 0, no IgκC/CD38-positive infiltrate; 1+, weak IgκC/CD38-positive infiltrate; 2+, moderate IgκC/CD38-positive infiltrate; and 3+, strong IgκC/CD38-positive infiltrate. To dichotomize the patients, tumor slides with a IgκC/CD38 score 0 and 1+ were considered as low plasma cell infiltration, whereas cases with 2+ and 3+ were high plasma cell infiltration, respectively.

### 4.4. Statistical Analysis

Statistical analyses were performed using the SPSS statistical software program, version 28.0 (IBM SPSS Statistics for Windows, Version 28.0., Released 2021 NY, USA, IBM Corp.

Patients’ characteristics were given in absolute and relative numbers. The prognostic significance of the immunohistochemically determined the IgκC and CD38 expression for MFS was examined via the Kaplan–Meier survival analysis as well as the univariate and multivariate Cox Regression analysis adjusted for age (≤50 vs. >50 years), pT-stage (≤2 vs. >2 cm), histological grade (GI + GII vs. GIII), and lymph node status (negative vs. positive). The significance of Kaplan–Meier survival analysis was assessed using the *p* value of the Log-Rank test. A possible correlation between the strength of plasma cell infiltration on the one hand and the histological degree of differentiation or the proliferation index Ki67 on the other hand was calculated using ordinal regression. The Spearman-Rho correlation coefficient was used to assess whether there was a correlation effect between the IgκC and C38 expression as well as a possible correlation between the intensity of plasma cell infiltration and the presence of a BRCA mutation (at the germline level).

### 4.5. Validation of Immunohistochemical Results Using mRNA Expression Data of IgκC and CD38 of an Independent TNBC Cohort

To validate our immunohistochemical results concerning the prognostic impact of the plasma cell markers IgκC and CD38 in a larger, independent cohort of TNBC, we used the publicly available gene expression data of IgκC and CD38 with their associated tumor characteristics, as well as the follow-up data [29] in a cohort of 424 TNBC samples.

## 5. Conclusions

In the present immunohistochemical retrospective study, we were able to show the prognostic significance of the humoral immune response in the observed collective of 107 patients with early triple-negative breast cancer. A higher number of CD38-positive tumor-infiltrating plasma cells was associated with a significantly longer MFS and BRCA mutations. Furthermore, CD38 was identified as an independent prognostic factor via the multivariate Cox regression analysis.

## Figures and Tables

**Figure 1 ijms-24-15219-f001:**
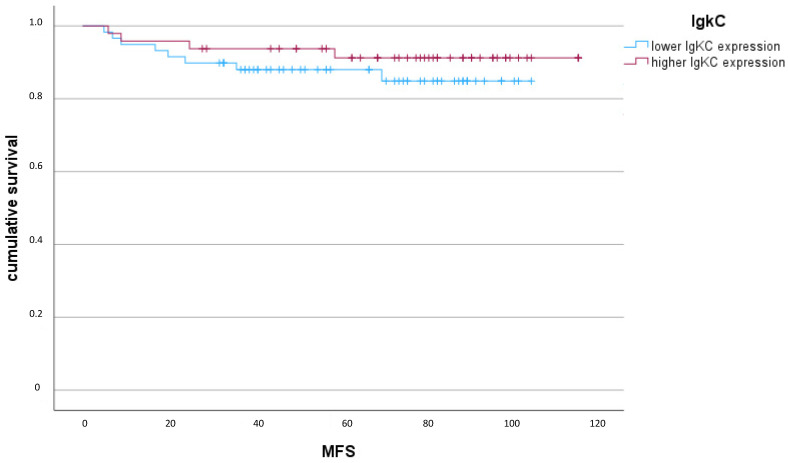
Kaplan–Meier Diagram of IgκC expression (*p* = 0.35, Log Rank) in terms of MFS (metastasis-free survival) (months).

**Figure 2 ijms-24-15219-f002:**
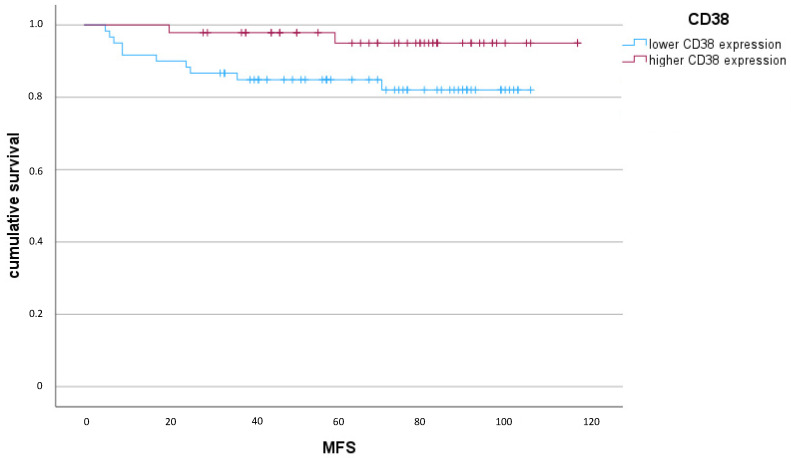
Kaplan–Meier Diagram of CD38 expression (*p* = 0.039 Log Rank) in terms of MFS (metastasis-free survival) (months).

**Figure 3 ijms-24-15219-f003:**
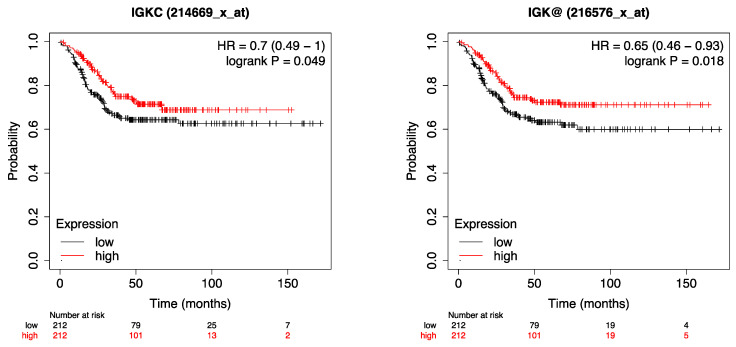
Kaplan–Meier plots of IgκC mRNA expression (probe set 214669_x_at and 216576_x_at) of an independent TNBC cohort showing IgκC as a positive prognostic marker in terms of MFS.

**Figure 4 ijms-24-15219-f004:**
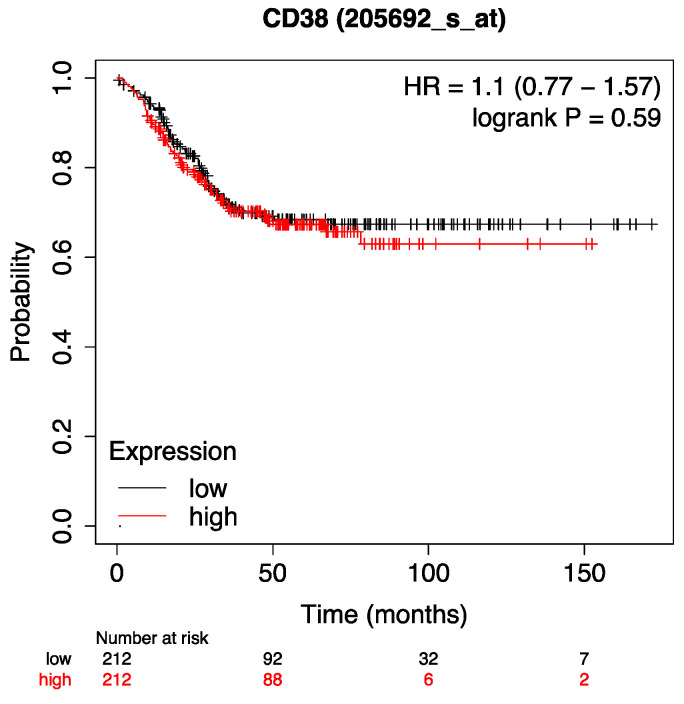
Kaplan–Meier plot of CD38 mRNA expression of an independent TNBC cohort: CD38 mRNA expression has no significant impact on MFS.

**Figure 5 ijms-24-15219-f005:**
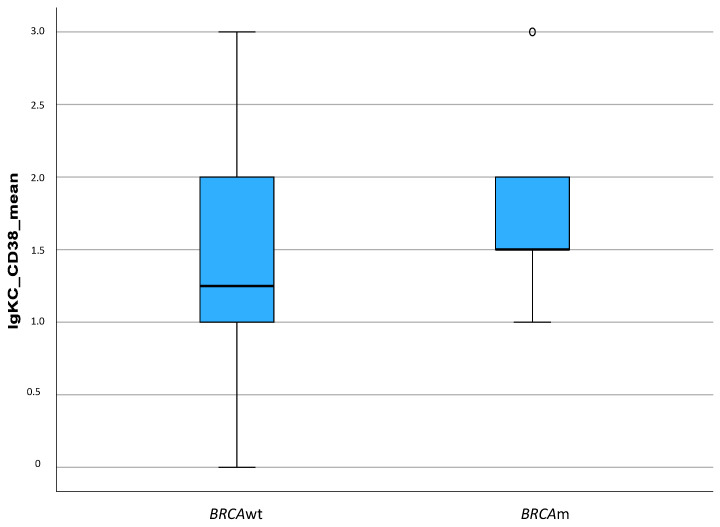
Box Plot Diagram showing the correlation between BRCA mutation (BRCAm) and the extent of plasma cell infiltration (r = 0.211, *p* = 0.029).

**Table 1 ijms-24-15219-t001:** Univariate Cox Regression Analysis in regard of MFS (metastasis-free survival).

	Multivariate Cox Regression MFS		
Variable	Hazard Ratio	95% CI	*p*-Value
**IgκC** Higher expression vs. lower expression	0.910	0.406–2.035	0.818
**Age** >50/=50 vs. <50	0.537	0.160–1.802	0.314
**Tumor Size** pT2–4 vs. pT1	2.219	0.537–9.168	0.271
**Lymph node status** pN1–3 vs. pN0	4.756	1.216–18.601	*0.025*
**Grading** G 1–2 vs. G 3	0.494	0.143–1.712	0.266

**Table 2 ijms-24-15219-t002:** Multivariate Cox Regressions analysis of IgκC in terms of MFS (metastasis-free survival) adjusted for age, tumor size, lymph node status, and tumor grade of differentiation.

	Univariate Cox Regression MFS		
Variable	Hazard Ratio	95% CI	*p*-Value
**IgκC** Higher expression vs. lower expression	0.852	0.415–1.751	0.664
**CD38** Higher expression vs. lower expression	0.508	0.252–1.027	0.059
**age** >50/= 50 vs. <50	0.677	0.218–2.099	0.499
**Tumor size** pT2–4 vs. pT1	2.641	0.715–9.758	0.145
**Lymph node status** pN1–3 vs. pN0	5.572	1.508–20.591	*0.010*
**Grading** G 1–2 vs. G 3	0.497	0.158–1.566	0.233

**Table 3 ijms-24-15219-t003:** Multivariate Cox Regressions analysis of CD38 in terms of MFS (metastasis-free survival) adjusted for age, tumor size, lymph node status, and tumor grade of differentiation.

	Multivariate Cox Regression MFS		
Variable	Hazard Ratio	95% CI	*p*-Value
**CD38** Higher expression vs. lower expression	0.438	0.195–0.983	*0.045*
**age** >50/= 50 vs. <50	0.496	0.152–1.619	0.245
**Tumor size** pT2–4 vs. pT1	1.869	0.459–7.603	0.382
**Lymph node status** pN1–3 vs. pN0	6.662	1.639–27.073	*0.008*
**Grading** G 1–2 vs. G 3	0.875	0.228–3.366	0.846

**Table 4 ijms-24-15219-t004:** Patients’ characteristics (in absolute and relative numbers).

	Number	Percentage
**Age at diagnosis**		
<50	44	41%
>50	63	59%
**Tumor size**		
pT1	49	46%
pT2–4	58	54%
**Lymph node status**		
pN0	68	64%
pN1–3	39	36%
**Tumor grade**		
G 1–2	28	26%
G 3	79	74%
**Ki67%**		
≤20%	7	6.5%
>20%	54	50.5%
Missing value	46	43%
**IgκC expression**		
weak (Score 0/1)	59	55.1%
strong (Score 2/3)	48	44.9%
**CD38 expression**		
weak (Score 0/1)	60	56.1%
strong (Score 2/3)	47	43.9%
**Chemotherapy**		
Yes	79	73.8%
No	28	26.2%
**BRCA Mutation**		
BRCAwt (wildtype)	90	84.1%
BRCAm (mutation)	17	15.9%

## Data Availability

Not applicable.

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
