# Peer review of "Prognostic Impact of CD38- and IgκC-Positive Tumor-Infiltrating Plasma Cells in Triple-Negative Breast Cancer"

_ijms, 2023, doi:10.3390/ijms242015219_

Round 1

Reviewer 1 Report

The authors studied a cohort of 107 TNBC with a median follow up of 5 years. They analyzed two markers for plasma cell, CD38 and IGKC (while it is not fully clear if data for both markers were available for all of the samples).

About half of the patients showed strong expression of the markers and were associated with improved prognosis.

Suggestions:

The layout and order of the tables (Tab2 vs Tab1) needs to be corrected.

Discussion section:

Line 311: "Recurrence" is expected to include distant metastasis. Did the authors mean Local Recurrence?

No results for stromal TIL scoring are presented for the analyzed cohort. Therefore, among the limitations of the study it may be mentioned, that the data so far do not provide evidence whether the specific detection of plasma cells delivers additional prognostic information as compared to stromal TILs.

Reviewer 2 Report

In the current study, authors have analyzed the prognostic value of CD38 and IgкC positive tumor infiltrating plasma cells in triple-negative breast cancer. The manuscript is well-written but I have the following comments

1- In the discussion section, could the author comment on the potential employment of the studied markers in a pan-cancer model? Is there any published study that analyzed the prognostic value of CD38 and IgкC positive tumor infiltrating plasma cells in human tumors other than breast cancer?

2- In line 18, a sentence started with a number, kindly edit.

3- In line 51, the symbol (:) was added in the middle of a sentence. Is that a typo?

4- The presentation of tables 1 and 2 needs editing. The table title should always come before inserting the table.

Minor editing of English language required

Round 2

Reviewer 2 Report

The manuscript can be accepted in the current format.

Minor editing of English language required